# Analysis of Novel Immunological Biomarkers Related to Rheumatoid Arthritis Disease Severity

**DOI:** 10.3390/ijms241512351

**Published:** 2023-08-02

**Authors:** Sandra Pascual-García, Pascual Martínez-Peinado, Ana B. López-Jaén, Francisco J. Navarro-Blasco, Yoel G. Montoyo-Pujol, Enrique Roche, Gloria Peiró, José M. Sempere-Ortells

**Affiliations:** 1Department of Biotechnology, University of Alicante, 03690 San Vicente del Raspeig, Spain; 2Rheumatology Unit, University General Hospital of Elche, 03203 Elche, Spain; 3Medical Oncology Department, Dr. Balmis University General Hospital, Pintor Baeza 12, 03010 Alicante, Spain; 4Alicante Institute for Health and Biomedical Research (ISABIAL), Pintor Baeza 12, 03010 Alicante, Spain; 5Biochemistry and Cell Therapy Unit, Institute of Bioengineering, Miguel Hernandez University of Elche, 03202 Elche, Spain; 6Pathology Department, Dr. Balmis University General Hospital, Pintor Baeza 12, 03010 Alicante, Spain

**Keywords:** rheumatoid arthritis, biomarkers, disease severity, leukocyte phenotype, B lymphocytes, cytotoxic T lymphocytes, NK/T lymphocyte ratio, effector memory cells, central memory cells

## Abstract

Rheumatoid factor (RF) and anti-citrullinated protein antibodies (ACPAs) are the most frequently used rheumatoid arthritis (RA) diagnostic markers, but they are unable to anticipate the patient’s evolution or response to treatment. The aim of this study was to identify possible severity biomarkers to predict an upcoming flare-up or remission period. To address this objective, sera and anticoagulated blood samples were collected from healthy controls (HCs; n = 39) and from early RA (n = 10), flare-up (n = 5), and remission (n = 16) patients. We analyzed leukocyte phenotype markers, regulatory T cells, cell proliferation, and cytokine profiles. Flare-up patients showed increased percentages of cluster of differentiation (CD)3^+^CD4^−^ lymphocytes (*p* < 0.01) and granulocytes (*p* < 0.05) but a decreased natural killer (NK)/T lymphocyte ratio (*p* < 0.05). Analysis of leukocyte markers by principal component analysis (PCA) and receiver operating characteristic (ROC) curves showed that CD45RO^+^ (*p* < 0.0001) and CD45RA^+^ (*p* < 0.0001) B lymphocyte expression can discriminate between HCs and early RA patients, while CD3^+^CD4^−^ lymphocyte percentage (*p* < 0.0424) and CD45RA^+^ (*p* < 0.0424), CD62L^+^ (*p* < 0.0284), and CD11a^+^ (*p* < 0.0185) B lymphocyte expression can differentiate between flare-up and RA remission subjects. Thus, the combined study of these leukocyte surface markers could have potential as disease severity biomarkers for RA, whose fluctuations could be related to the development of the characteristic pro-inflammatory environment.

## 1. Introduction

Rheumatoid arthritis (RA) is a systemic, autoimmune disease that affects more women than men [1] and involves inflammation of both the synovia and connective tissue of joints, although it can also involve extra-articular symptoms such as depression, changes in the lipid profile, and many others [2,3]. The cause of the disease is not known, although certain factors may have an important role, such as genetics, epigenetics, the presence of anti-citrullinated protein antibodies (ACPAs) and rheumatoid factor (RF), the microbiome, and the immune system [4,5,6,7,8]. Even though ACPAs and RF have been used as biomarkers to diagnose RA, their capability to predict the prognosis or therapeutic response of patients has not been demonstrated [9]. 

This disease is characterized by the presence of a pro-inflammatory environment due to cytokines such as tumor necrosis factor (TNF)-α or interleukin (IL)-6 [10] and also by a decrease in regulatory T cells (Tregs) [8]. In addition, some proteins have an important role in the generation of a pro-/anti-inflammatory environment. This is the case with endothelial protein C receptor (EPCR), which is part of the EPCR-activated protein C (APC)-protease activated receptor (PAR1) system and exhibits a marked anti-inflammatory and immunosuppressive ability [11,12]. However, some authors have described the pro-inflammatory role of this receptor in RA [13]. In addition, protein C (PC) is, among others, a ligand of the epidermal growth factor receptor (EGFR), whose serum levels are highly increased in this disease [14]. IgG fragment crystallizable (Fc) receptors (cluster of differentiation (CD)16, CD32, and CD64) have also been studied in RA patients. CD16 has a key role in antibody-dependent cytotoxicity, and its level of expression, together with the analysis of CD14, can divide the monocytes into different subpopulations [15]. In RA patients, disease activity score (DAS) 28 levels have been correlated with an increase in CD64 expression in classical and intermediate monocytes [15]. CD45RA and CD45RO are two important molecules in the study of memory and naïve cells. Their expression, together with CD62L selectin in cytotoxic T lymphocytes (CTLs), has also been used to determine the transition from naïve to effector memory cells [16,17], while CD11a antigen, a subunit of the integrin lymphocyte function-associated antigen 1 (LFA-1), has been described to have a central role in leukocyte migration [18].

Other molecules that are being studied as potential biomarkers for several diseases are miRNAs (miR) [19,20]. Recent studies performed in RA patients have discovered that an increased expression of some miRNAs, such as let-7d, miR-24, miR-126, miR-130a, miR-221, or miR-431, could be predictive as a biomarker for this disease [21]. Moreover, prior studies in these patients have found that some miRNAs have either a pro-inflammatory or a regulatory role, e.g., miR-155 and miR-146a, respectively [22,23]. Other miRNAs have been described to have both functions, e.g., miR-21 [24,25,26].

In this study, we characterized diverse leukocyte populations, cytokine profiles, and exosomal miRNAs in samples obtained from patients with RA, looking for potential clinical biomarkers for this disease. Additionally, the use of routine and accessible techniques for such characterization, like peripheral blood analysis, could facilitate the standardization of the clinical use of these biomarkers. Our hypothesis posits that some of these immunological biomarkers could serve as indicators for an impending relapse or transition into a remission stage, which could be used to improve disease management.

## 2. Results

### 2.1. Analysis of the Recruited Patients

For this study, 31 RA patients and 39 healthy controls (HCs) were recruited. RA patients were treated with synthetic and/or biologic disease-modifying antirheumatic drugs (DMARDs) (Table 1). 

Early RA (ERA) vs. remission patients showed significant differences in painful and swollen joints (*p* < 0.0001, *p* < 0.001, respectively), level of discomfort and pain (*p* < 0.01, *p* < 0.001, respectively), evaluation of the symptomatology by the rheumatologist (*p* < 0.001), serum C reactive protein (CRP) levels (*p* < 0.01), and DAS28 (*p* < 0.0001). On the other hand, significant differences between RA flare-up and remission patients were also found in the number of painful and swollen joints (*p* < 0.01, *p* < 0.01, respectively), level of discomfort and pain (*p* < 0.001, *p* < 0.001, respectively), evaluation of RA symptoms by the rheumatologist (*p* < 0.001), serum CRP levels (*p* < 0.01), and DAS28 (*p* < 0.0001) (Table 2).

### 2.2. Leukocyte Populations Phenotyping

Lymphocyte subpopulations, monocytes, and granulocytes were assessed following a flow cytometry gating strategy (Figure 1). Regarding total gated lymphocytes, we found a decrease in RA subjects (23.02%), especially in ERA (23.36%), flare-up (20.32%), and remission patients (24.6%) compared to HCs (34.4%; *p* < 0.0001, *p* < 0.01, *p* < 0.01, *p* < 0.01, respectively) (Figure 2A, Appendix A). When lymphocyte subpopulations were analyzed, although no significant differences were found between RA patients and healthy volunteers, lower percentages were found in remission subjects vs. ERA (*p* < 0.05) and flare-up (*p* < 0.01) of both T (CD3^+^) and CD3^+^CD4^−^ lymphocytes (Figure 2B,C). Regarding this latter lymphocyte subpopulation, flare-up patients showed higher percentages (36.56%) compared to ERA ones (27.94%, *p* < 0.05; Figure 2C). In addition, all study subjects showed similar values of T helper (Th; CD3^+^CD4^+^) lymphocytes, but flare-up patients presented a slight decrease (37.2%; Figure 2D). Natural killer (NK; CD3^−^CD16^+^CD56^+^) lymphocytes (Figure 2E) decreased in all RA patients (12.99%) compared to HCs (15.7%), but the subgroups showed different patterns: ERA and flare-up (*p* < 0.05) subjects showed a reduction (11.26% and 8.46%, respectively), while remission patients presented a minor increase for these cells (17.96%). This pattern was also observed with the NK/T lymphocyte ratios (Figure 2F) between RA and HCs subjects (0.22% and 0.27%, respectively), with lower values observed in flare-up patients (0.11%, *p* < 0.05), while remission ones showed a slight increase (0.35%, *p* < 0.05). B lymphocyte (CD3^−^CD19^+^) values were very similar between all subjects (Figure 2G). On the other hand, despite the similarity in monocyte values between HCs and RA subjects (6.13% and 6.31%, respectively), we observed a slight elevation in patients experiencing flare-ups (6.92%), although the differences were not statistically significant (Figure 2H). Finally, granulocyte percentages were increased in RA patients compared to healthy volunteers (55.77% and 46.7%, respectively; *p* < 0.05); in particular, flare-up patients presented a dramatic increase in their percentage (61.72%, *p* < 0.05; Figure 2I).

### 2.3. Analysis of Biomarkers Expression by PCA

Percentages of different leukocyte subpopulations expressing EGFR, EPCR, CD16, CD32, CD64, CD45RO, CD45RA, CD62L, and CD11a were able to distinguish two distinctive groups: HCs and RA patients. The former had higher CD45RA and CD11a expression, while CD45RO expression was positively related to having RA (Figure 3A–H). This relationship was inverted and even stronger in B lymphocytes, where RA patients showed a strong association with CD45RA expression (Figure 3F).

Principal component analysis (PCA) and correlation matrix analyses of biomarker expression percentages allowed the detection of differential expression patterns between different leukocyte subpopulations of RA patients and HCs. Regarding the total, T, Th, and CD3^+^CD4^−^ lymphocyte populations, a decrease in EGFR expression was observed in RA patients (6.44%, 5.73%, 5.62%, and 5.59%, respectively) with respect to HCs (31.20%; Figure 3A–D; Appendix A). Regarding CD3^+^CD4^−^ lymphocytes, a decrease in CD11a expression was observed in patients (60.73%) compared to HCs (99.33%), as well as a reduction in this integrin expression in ERA and remission subjects in Th cells (23.16% and 35.14%, respectively). If we analyze CD45RA and CD45RO expression on CD3^+^CD4^−^ lymphocytes, two opposing patterns are observed. Expression levels for CD45RA were elevated in HCs and remained similar with respect to RA patients in flare-up (90.6% and 84.1%, respectively), whereas they decreased in the remaining subjects. On the other hand, CD45RO expression was elevated only in ERA and remission patients (88.11% and 89.17%, respectively) compared to healthy volunteers (43.94%). This pattern was maintained in NK (Figure 3E) and B lymphocytes (Figure 3F). In NK cells, decreased CD62L values were observed in patients vs. HCs (39.65% and 74.42%, respectively), especially in ERA and remission subjects (38.01% and 29.11%, respectively). However, in B lymphocytes, the pattern was reversed, as all the analyzed samples expressed similar values, except for patients in flare-up, who experienced a dramatic decrease in CD62L values (26.7%). In addition, in this lymphocyte population, CD11a expression was similar among RA patients and HCs (2.71%), except in flare-up subjects, who showed a marked increase (75.66%). In the case of monocytes (Figure 3G), EGFR and CD11a expression in patients decreased (22.72% and 61.51%, respectively), while CD32 levels increased (88.29%) compared to healthy volunteers (53.78%, 97.62%, and 42.98%, respectively). Finally, granulocytes (Figure 3H) showed a similar expression pattern for the EGFR, EPCR, and CD11a markers (28.56%, 17.17%, and 32.04%, respectively), whose levels decreased in patients compared to HCs (96.63%, 47.33%, and 97.65%, respectively). However, CD11a expression values in HCs were similar to those in patients in flare-up (78.7%). Regarding CD62L selectin, most RA subgroups and HCs showed similar values (80.34% and 99.23%, respectively), but flare-up patients experienced a decrease in this antigen (32.94%). No significant changes were found for CD16 and CD64.

From all the markers tested, we selected a few that might be ideal candidates for disease severity biomarkers, such as the percentages of CD3^+^CD4^−^ lymphocytes, NK cells, and granulocytes, the NK/T lymphocyte ratio, and the percentages of CD45RO^+^, CD45RA^+^, CD62L^+^, and CD11a^+^ B lymphocytes, to perform receiver operating characteristic (ROC) curve analysis. When comparing HCs vs. ERA patients, only the expression of CD45RO (*p* < 0.0001, area under the curve (AUC) = 1) and CD45RA (*p* < 0.0001, AUC = 1) in B lymphocytes was able to fully discriminate between subjects (Figure 4A). Other markers that showed the ability to differentiate between flare-up and remission patients were CD3^+^CD4^−^ lymphocytes (*p* = 0.0424, AUC = 0.8571) and the expression of CD45RA (*p* = 0.0424, AUC = 0.8571), CD62L (*p* = 0.0284, AUC = 0.8857), and CD11a (*p* = 0.0185, AUC = 0.9143) in B lymphocytes (Figure 4B). These markers presented similar capacities to discriminate between these two types of patients as compared to traditional diagnostic markers, such as RF (*p* = 0.0475, AUC = 0.5875), CRP (*p* = 0.0039, AUC = 0.9375), and DAS28 (*p* = 0.0010, AUC = 1). Although the percentage of NK cells (*p* = 0.0618, AUC = 0.8286), the NK/T lymphocyte ratio (*p* = 0.0618, AUC = 0.8286), and the expression of CD45RO (*p* = 0.0618, AUC = 0.8286) in B lymphocytes did not reach statistically significant differences, they might be able to distinguish between patients in flare-up and in remission with lower sensitivity and specificity. However, due to the values obtained for CD3^+^CD4^−^ lymphocytes (*p* = 0.5072, AUC = 0.5850), NK cells (*p* = 0.2125, AUC = 0.6599), granulocyte percentages (*p* = 0.1000, AUC = 0.7109), the NK/T ratio (*p* = 0.2540, AUC = 0.6463), and the expression of CD62L (*p* = 0.4381, AUC = 0.6048) and CD11a (*p* = 0.3067, AUC = 0.6381) in B lymphocytes, these measures would not be able to discriminate between healthy donors and RA patients. The percentage of granulocytes (*p* = 0.1675, AUC = 0.7429) and ACPA (*p* = 0.5633, AUC = 0.5875) also failed to differentiate between the flare-up and remission stages. 

### 2.4. Treg Cell Population Analysis

Patients with RA showed a decrease in the values of CD4^+^CD25^+^FoxP3^+^ Treg cells (0.4464%) compared to HCs (2.034%, *p* < 0.001; Appendix A; Appendix A). Adalimumab addition to the cultures of RA patients’ PBMCs slightly enhanced the decrease in CD4^+^CD25^+^FoxP3^+^ Treg percentage (*p* < 0.0001; Appendix A). Stimulation with PHA caused an increase in Treg cells in the cultures, particularly in RA patients, which reached values similar to those of the HCs (6.529%) but with no significant differences (Appendix A).

### 2.5. Analysis of Lymphocyte Proliferation

RA patients showed significantly lower values of proliferation than did HCs under both unstimulated and stimulated conditions (*p* < 0.0001; Appendix A; Appendix A). The addition of adalimumab to PBMC cultures from RA patients resulted in a slight decrease under both unstimulated (*p* = 0.0507; Appendix A) and stimulated conditions (*p* = 0.086; Appendix A).

### 2.6. Serum Th1/Th2/Th17 Cytokine Profile

An analysis of serum cytokines showed a typical pro-inflammatory environment in RA patients vs. HCs, mediated by increased levels of IL-17A, TNF-α, IL-6, and IL-2. Compared to HCs, remission patients presented increased concentrations of TNF-α (46.4 pg/mL, *p* < 0.05), IL-17A (55.8 pg/mL), and IL-10 (15 pg/mL, *p* < 0.05). Regarding ERA patients, higher values of TNF-α (36.5 pg/mL) and lower levels of IL-17A (19.6 pg/mL) were found compared to HCs (11.2 pg/mL, *p* < 0.01) and remission patients (55.8 pg/mL, *p* < 0.05), respectively. Finally, remission patients showed higher values of this latter cytokine when compared to flare-up patients (20.4 pg/mL, *p* < 0.05) (Table 3).

### 2.7. Supernatant Th1/Th2/Th17 Cytokine Profile

In the supernatant from the PBMC cultures under non-stimulated conditions, cytokine levels were below or near the level of detection for all conditions, with no significant differences. The addition of adalimumab to the culture did not produce any remarkable changes (Table 4A). On the other hand, under stimulated conditions, RA patients showed statistically significant lower levels of interferon (IFN)-γ (276.9 pg/mL, *p* < 0.05) and TNF-α (109.9 pg/mL, *p* < 0.05) when compared to HCs (1115 pg/mL and 358.9 pg/mL, respectively). These levels declined further with the addition of adalimumab, reaching significant differences in the cases of IFN-γ (54.6 pg/mL, *p* < 0.01), TNF-α (3.9 pg/mL, *p* < 0.001), and IL-10 (328.9 pg/mL, *p* < 0.01) compared to HCs. In addition, the IL-10 level was also lower than that of RA subjects (1021 pg/mL, *p* < 0.05). Furthermore, RA patients showed higher levels of IL-6 when compared to HCs (15537 pg/mL and 6839 pg/mL, respectively), which were reduced to normal values with the addition of adalimumab to the culture (5337 pg/mL, Table 4B). 

### 2.8. Serum Exosomal miRNA Analysis

Exosomes isolated from serum samples from the HCs and RA patients showed elevated expression of CD63 and CD9, regardless of their disease stage (Appendix A). In terms of exosomal miRNAs, the serum samples from RA patients showed increased expression of miR-21, especially in flare-up subjects, followed by an increase in miR-155 in remission patients, where miR-146a also achieved its maximum level (Figure 5; Appendix A).

## 3. Discussion

In this article, we aimed to identify RA disease severity biomarkers; therefore, we characterized leukocyte populations by examining their membrane expression of EGFR, EPCR, IgG Fc receptors, adhesion molecules, and cell activation markers. We also assessed Treg cells, lymphocyte proliferation, and the profile of Th1/Th2/Th17 cytokines in patient serum and PBMC supernatant samples. We also conducted ex vivo assays to examine the impact of adalimumab (Humira^®^), an anti-TNF-α antibody, and the expression of exosomal miRNAs in serum samples obtained from RA patients. This study showed that some of the analyzed leukocyte parameters and expression antigens could be used as potential disease severity biomarkers since they were fully capable of discriminating between HCs and ERA patients (e.g., CD45RO^+^ and CD45RA^+^ B lymphocytes). On the other hand, CD3^+^CD4^−^ lymphocytes and CD45RA^+^, CD62L^+^, and CD11a^+^ B lymphocytes were able to differentiate between RA patients experiencing a flare-up or a remission of their symptoms.

The analysis of clinical data from RA patients gave us an insight into the degree of effectiveness of the treatment applied in every case. In this sense, it was observed that the number of painful and swollen joints, the level of discomfort and pain, the evolution of the severity of their symptoms according to the rheumatologist, and their CRP, RF, ACPA, and DAS28 values were reduced once biological DMARDs were given to the patients (vs. methotrexate alone), as usually happens in remission. Therefore, as recommended by the European League Against Rheumatism (EULAR) [27], we suggest that early treatment with biologics could be an interesting option for ERA patients whenever possible in order to bring forward their clinical improvement.

When looking at changes in leukocyte values, a decrease in the number of total lymphocytes was found, regardless of disease status, compared to HCs, as previously described [28]. When looking at lymphocyte subpopulations, we observed a significant elevation of CD3^+^CD4^−^ lymphocytes in flare-up patients, which, according to our previous studies, could be related to their increased RA activity [8]. Although the CD3^+^CD4^−^ population can contain some double-negative T lymphocytes and very small percentages of different CD8^+^ regulatory subsets, most of this population is usually represented by CD3^+^CD8^+^ T lymphocytes that become cytotoxic after activation. In addition, the increased presence of CD3^+^CD4^−^ lymphocytes in these patients has also been associated with higher DAS28 values and pro-inflammatory cytokine production [8,29], which highlights their leading role in the exacerbation of symptoms in RA patients. The analysis of ROC curves reinforced CD3^+^CD4^−^ lymphocyte percentage as a strong candidate biomarker for discriminating between flare-up and remission patients. However, NK cells from RA patients followed the opposite pattern, which could be related to their potential regulatory function [30,31]. In fact, the NK/T cell ratio is increasingly gaining acceptance as a novel marker that has already been demonstrated to be able to predict disease status in other autoimmune diseases, such as multiple sclerosis [32,33]. The observed decrease in this ratio in our study would therefore suggest a worsening of symptoms; therefore, we postulate that changes in the NK/T cell ratio could also be used as a predictive biomarker of an upcoming remission or relapse period in RA patients. Another subpopulation that also increased in flare-up patients was granulocytes, composed mostly of neutrophils, which, as previously described, contribute to the symptomatology observed in these patients due to the ability of these cells to infiltrate joints [34,35]. 

The multi-variant PCA analysis carried out on the different cell phenotypes showed two different clusters for HCs and RA patients in all the leukocyte populations analyzed, which were very distinct in the case of B lymphocytes. The results of this analysis, together with the matrix plot, showed decreased EGFR expression in several leukocyte subpopulations (total, T, Th, and CD3^+^CD4^−^ lymphocytes, monocytes, and granulocytes) of RA patients. This finding is in line with other studies that link this inhibition to the increase in some miRNAs, such as miR-133, which would hinder the viability and migration of pathogenic cells involved in the onset of RA [36]. Other authors have described an EGFR increase in RA patients, but in that study, the molecule was measured in serum and not on the cell surface [14]. We also analyzed the expression of EPCR, which, according to previous studies, could be playing a pro-inflammatory role in this disease [13]. The decreased expression of EPCR in granulocytes from RA patients could indicate the existence of a pro-inflammatory environment leading to the onset of this disease, as described in an experimental model of colitis [37]. Regarding our results on the expression of the FcγRII receptor (CD32), we observed an increased expression in monocytes from RA patients, as previously described in the literature [38,39]. Although CD32 has different isoforms with pro- or anti-inflammatory effects, it has been observed that cells involved in the pathogenesis of RA express the isoform with constitutive immunoreceptor tyrosine-based activation motifs (ITAM), which would contribute to a pro-inflammatory environment suitable for the development of this disease [40]. Previous studies have defined different lymphocyte subpopulations according to the expression of CD45RA, CD45RO, and the CD62L antigens: central memory cells (CD45RA^−^CD45RO^+^CD62L^+^), effector memory cells (CD45RO^+^CD45RA^−^CD62L^−^), and effector cells (CD45RO^−^CD45RA^+^CD62L^−^) [16,17]. Our results show that most lymphocytes from flare-up patients might correspond to effector phenotypes (CD62L^−^), which could be related to their more severe symptomatology. However, ERA and remission patients showed a phenotype closer to the central memory one, which would start to exert effector functions after receiving the appropriate stimulus. This transition from a naïve/central memory to an effector phenotype can be observed mainly in B lymphocytes and in a more diffuse manner in the rest of lymphocyte populations, thus indicating a potential role of these antigens as biomarkers of disease stage. In fact, the ROC curves showed that the expression of CD45RO in B lymphocytes was able to discriminate between HCs and ERA patients, while the expression of CD45RA in this lymphocyte population could differentiate between healthy volunteers and ERA, as well as between flare-up and remission patients; therefore, these proteins could be ideal candidates for disease severity biomarkers. Finally, CD11a antigen was highly expressed in most leukocyte populations from flare-up patients; it was mainly visible on B lymphocytes while decreasing in monocytes of patients regardless of their disease status. The B lymphocyte CD11a result correlates with the ROC curves, as this parameter was capable of distinguishing between flare-up and remission patients. According to previous studies [41], the increased expression of this membrane antigen on B lymphocytes in flare-up patients also suggests an effector phenotype.

Regarding Treg cells, the significantly lower values detected in RA patients compared to HCs agree with what has already been described by other authors [8,42]. Additionally, PBMC proliferation also decreased in these patients, particularly after the addition of adalimumab, reaffirming the anti-proliferative role of this drug [43]. 

The analysis of cytokine profiles of serum samples from patients confirmed a general imbalance towards a predominant pro-inflammatory environment compared to the HCs, as previously published, which is probably related to the different degrees of bone and cartilage erosion that are typical of this disease [10,44,45]. It is worth noting the fact that remission patients presented increased levels of some pro-inflammatory cytokines such as IL-6, TNF-α, and IL-17A, even at higher levels than those of the flare-up patients. However, these patients were also the ones presenting the highest levels of IL-10. An increased production of IL-17A by peripheral Th cells after TNF blockade in RA patients has been previously related to an inhibition of the expression of migration-associated chemokine receptors, which could prevent the arrival of these potentially pro-inflammatory cells at the synovium [46,47]. Therefore, although remission patients seem to maintain a pro-inflammatory environment, the simultaneous increase in cytokines, such as IL-10 or IL-4, in these patients could imply the existence of an anti-inflammatory compensatory mechanism that might be related to the effectiveness of the administered treatment. On the other hand, the stimulated PBMCs released cytokines, showing that IL-6 seems to be the most expressed cytokine, thus notably contributing to the generation of the pro-inflammatory environment and clinical symptoms of these patients [45]. The addition of adalimumab to the culture decreased the concentrations of all the studied cytokines, reaching significant values for IFN-*γ*, TNF-α, and IL-10 compared to the HCs and for this latter cytokine compared to untreated PBMCs from RA patients. This could be an indication that the final mechanism of action of these biological drugs would be carried out not only by a direct blockade of the targeted cytokine but also through an additional indirect immunomodulatory effect that would involve other cytokines, as previously described in the literature [48]. 

Finally, even though it has been described that miR-21 may have anti-inflammatory functions [24,25], our analysis of exosomal miRNAs in serum samples from RA patients showed an up-regulation of this miRNA, especially in flare-up patients, which, according to other authors, could play a pro-inflammatory role in this disease [26]. The higher expression of miR-155 (usually associated with pro-inflammatory functions [22]) in remission patients from our study is in line with the aforementioned increase in pro-inflammatory cytokines in the serum of these patients, therefore reinforcing the idea of a remaining basal inflammation in these patients. The slightly upregulated serum levels of miR-146a found in remission patients from our study would support a regulatory role for this miRNA in basal inflammation [23]. 

This study had a limitation in terms of the number of patients in the three groups, which could have an impact on the statistical analyses. Furthermore, we encountered a significant challenge in recruiting patients in the flare-up stage. This difficulty is probably due to the effectiveness of the biological DMARD treatments administered, which makes it harder to identify individuals with intensified symptoms. Since there is no unique standardized protocol for measuring certain surface markers on leukocytes, we may encounter inter-individual and inter-group variability of marker expression in patients. Finally, this study also has an instrumental limitation because of the combination of the multiple parameters used in the analysis. 

Nevertheless, we believe that the future application of these biomarkers in the clinic could potentially help to predict which patients may have the disease, as well as whether a patient in remission may experience an exacerbation of symptoms, and vice versa. All this would help improve patient management and quality of life. Further studies will be necessary to definitively confirm these data.

## 4. Materials and Methods

### 4.1. Participants

All procedures involving human cells were approved by the University of Alicante Ethics Committee and carried out in accordance with the principles of the Declaration of Helsinki. From 2016 to 2019, RA patients were randomly recruited from the General University Hospital of Elche and divided into ERA, with a disease duration of less than one year, and long-standing RA (LRA), with a disease duration longer than one year, who were distributed into two subgroups according to DAS28 status: flare-up (DAS28 > 2.6) and remission (DAS28 < 2.6) patients. Healthy volunteers were randomly recruited from the University of Alicante Medical Service. RA patients and HCs that arrived at General University Hospital of Elche and University of Alicante Medical Service consultations, respectively, signed a written informed consent form to take part in this study. The diagnosis of RA patients was performed using the American College of Rheumatology (ACR) and EULAR criteria [49]. Moreover, the specific inclusion and exclusion criteria were as follows: For RA patients, the inclusion criteria were: (1) individuals aged between 18 and 75 years; (2) both males and females; and (3) the absence of any other systemic disease. On the other hand, the exclusion criteria were: (1) individuals below 18 years of age or above 75 years of age; (2) the presence of other immune system diseases; and (3) those patients who could not adequately cooperate in the study. The control group consisted of individuals who were matched for age and sex, following these inclusion criteria: (1) individuals aged between 18 and 75 years; (2) both males and females; (3) no indications of joint pain or swelling; and (4) no inflammatory or other chronic diseases. For RA patients, their number of painful and swollen joints, CRP, RF, and ACPA serum levels, and DAS28 were measured. The level of discomfort and pain and the evaluation of RA symptoms by the rheumatologists were assessed using a metric scale. Healthy volunteers and RA patients’ samples were analyzed together.

### 4.2. Isolation and Culture of PBMCs

PBMCs were obtained through density gradient centrifugation in Lymphosep (Biowest, Nuaillé, France) and were cultured in complete Roswell Park Memorial Institute (RPMI) medium (Capricorn Scientific, Ebsdorfergrund, Germany) containing 10% fetal bovine serum (FBS) (Gibco, Boston, MA, USA), 1% penicillin/streptomycin (Hyclone, Logan, MA, USA), and 1% glutamine (Sigma-Aldrich, Saint Louis, MO, USA).

### 4.3. Membrane/Intracellular Leukocyte Analysis

Antigen expression in fresh whole blood was analyzed by flow cytometry (FACSCanto; BD Bioscience, Franklin Lakes, NJ, USA) using monoclonal antibodies against human CD16, CD19, CD11a, and CD45RA (BD Bioscience, Franklin Lakes, NJ, USA), EPCR and CD4 (BioLegend, San Diego, CA, USA), EGFR (Abcam, Cambridge, UK), CD3, CD56, CD32, CD64, CD45RO, and CD62L (eBioscience, Waltham, MA, USA). Treg analysis was performed using cultured PBMCs. For this purpose, monoclonal antibodies against human CD4, CD25, and FoxP3 (eBioscience, Waltham, MA, USA) and a fixation/permeabilization solution for FoxP3 intranuclear staining were used (eBioscience, Waltham, MA, USA). PCA and a color scale map for the analyzed antigens were performed/generated using Past 4 (v.4.07b, University of Oslo, Oslo, Norway). For PCA, an XY scatter plot with 95% confidence ellipses was generated to analyze the influence of the expression of each marker in the principal component. 

### 4.4. Proliferation Assay and Treg Analysis

To analyze lymphocyte proliferation, PBMCs were stained with 5 µM carboxyfluorescein succinimidyl ester (CFSE) (Sigma-Aldrich, Saint Louis, MO, USA), as previously described [50]. Briefly, 10^5^ PBMCs were cultured in flat-bottomed 96-well plates in complete RPMI. The culture was stimulated with 10 µg/mL of phytohaemagglutinin (PHA; Sigma, Saint Louis, MO, USA) and 8 µg/mL of Humira^®^ (AbbVie, North Chicago, IL, USA) and incubated at 37 °C and 5% CO_2_ for 120 h until flow cytometry analysis. Treg quantification was performed after 72 h of culture, and cells were separated from the supernatant by centrifugation at 360× *g*. Supernatants were stored at −20 °C until further cytokine study, and cells were stained with specific monoclonal antibodies (mentioned in Section 4.3) to study this lymphocyte population.

### 4.5. Analysis of Cytokines

Measurements of TNF-α, IFN-γ, IL-17A, IL-10, IL-6, IL-4, and IL-2 levels were performed using a “BD Cytometric Bead Array (CBA) Human Th1/Th2/Th17 Cytokine kit” (BD Bioscience, Franklin Lakes, NJ, USA) and analyzed by flow cytometry (FACSCanto; BD Bioscience, Franklin Lakes, NJ, USA), following the manufacturer’s instructions. Serum and PBMC culture supernatant cytokine concentrations (pg/mL) were obtained using FACSDiva software (v.8.0.1, BD Bioscience, Franklin Lakes, NJ, USA).

### 4.6. Isolation of Exosomes

Sera samples were first centrifuged for 45 min at 15,000× *g* and 4 °C, followed by 0.2 µm filtration of the supernatant and ultracentrifugation for 2 h at 100,000× *g* and 4 °C; the supernatant was then discarded. The isolation of exosomes was verified by flow cytometry.

#### Analysis of CD63 and CD9 Expression by Flow Cytometry 

The study of CD63 and CD9 expression in exosomes was performed using an ExoStep Cell Culture kit (Immunostep, Salamanca, Spain) and analyzed by flow cytometry (FACSCanto), following the manufacturer’s instructions. The percentage of CD63^+^CD9^+^ particles was obtained using Weasel software (v.3.0, Weasel, Australia).

### 4.7. Study of Exosomal miRNA Cargo

Exosomal RNA was extracted with TRIzol (Life Technologies, Carlsbad, CA, USA), chloroform (Honeywell, Charlotte, NC, USA), and a Directzol RNA Miniprep Plus kit (Zymo Research, Irvine, CA, USA), following the manufacturer’s instructions. RNA concentration and purity (260/280 ratio) were measured with a Nanodrop (Thermo Scientific, Waltham, MA, USA). Expression of miR-21, miR-146a, and miR-155 was measured by RT-qPCR (StepOnePlus; Thermo Fisher Scientific, Waltham, MA, USA), using miR-16 as a homogenous control due to its high stability, high abundance, and low variability [51,52,53,54,55]. A TaqMan MicroRNA Reverse Transcription Kit (Applied Biosystems, Waltham, MA, USA), Master Mix 2X ROX (NZYTech, Lisboa, Portugal), and the required primers for the RT-qPCR (Applied Biosystems, Waltham, MA, USA) were used following the manufacturer’s instructions. The RT-PCR program consisted of 1 cycle at 16 °C for 30 min, 1 cycle at 2 °C for 30 min, and 1 cycle at 85 °C for 5 min. The qPCR program followed a protocol of 1 cycle for preincubation at 95 °C for 10 min, 45 cycles for amplification at 95 °C for 15 s, 60 °C for 1 min, and 72 °C for 1 s, and, finally, 1 cycle for cooling at 40 °C for 30 s. For calculations, the mean expression values of miR-16 were subtracted from the mean values of the other miRNAs in all samples. The resulting values were then subtracted from the mean values of the HCs for each miRNA. Finally, the transformation Y = 2^−y^ was applied to obtain ΔΔCt values.

### 4.8. Statistics

Statistical analysis was carried out with GraphPad Prism 8 (v.8.0.1, GraphPad Software, Boston, MA, USA). The Shapiro–Wilk normality test for all the data was performed, and outliers were excluded. A Student’s *t* test and analysis of variance (ANOVA) with Fisher’s least significant difference (LSD) test were carried out for normal distributions. For non-parametric data, the Mann–Whitney test and Kruskal–Wallis test with an uncorrected Dunn’s test comparison were used. For ROC curves, the AUC was calculated with the Wilson/Brown method. To analyze differences between exosomal miRNAs from the sera samples, a 2-way ANOVA with a Tukey comparison test was carried out. It was considered a statistically significant difference if the *p*-value was less than 0.05.

## 5. Conclusions

In conclusion, our results show that the expression of CD45RO and CD45RA in B lymphocytes can discriminate between HCs and ERA patients, while CD3^+^CD4^−^ lymphocyte percentage and the expression of CD45RA, CD62L, and CD11a in B cells can differentiate flare-up from remission in RA patients. Finally, the marked presence of several pro-inflammatory cytokines and miR-155 in the serum shows that there is a basal inflammatory state in remission patients, which would be counterbalanced by anti-inflammatory cytokines, such as IL-10 or IL-4, and by miR-146a.

## Figures and Tables

**Figure 1 ijms-24-12351-f001:**
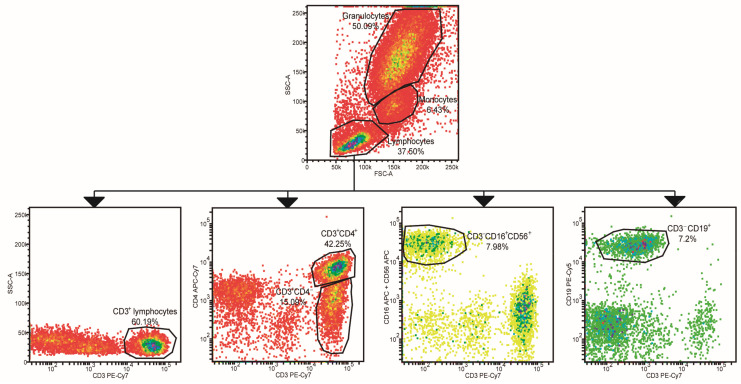
Leukocyte population gating strategy. Lymphocytes, monocytes, and granulocytes were gated based on size and granularity. Different lymphocyte subpopulations were identified according to cluster of differentiation (CD)3, CD4, CD16, CD56, and CD19 expression. Pseudo-colors in plots represent cell density.

**Figure 2 ijms-24-12351-f002:**
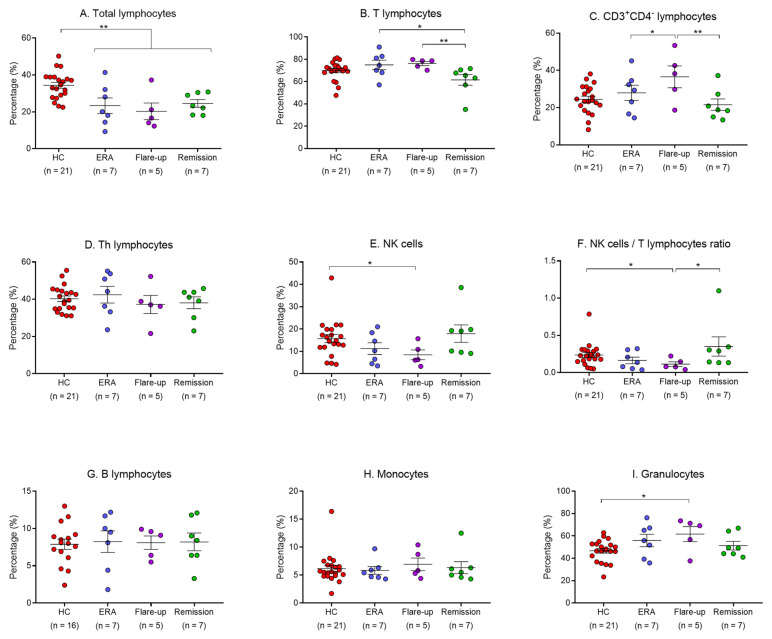
Analysis of leukocyte populations. Total lymphocytes (**A**), T lymphocytes (**B**), CD3^+^CD4^−^ lymphocytes (**C**), T helper (Th) lymphocytes (**D**), natural killer (NK) cells (**E**), NK cell/T lymphocyte ratio (**F**), B lymphocytes (**G**), monocytes (**H**), and granulocytes (**I**) were measured in HCs, ERA, flare-up, and remission patients. Bars show mean ± SEM. *p*-values were calculated using ANOVA (Fisher’s LSD test) and the Kruskal–Wallis test (uncorrected Dunn’s test). * = *p* < 0.05; ** = *p* < 0.01.

**Figure 3 ijms-24-12351-f003:**
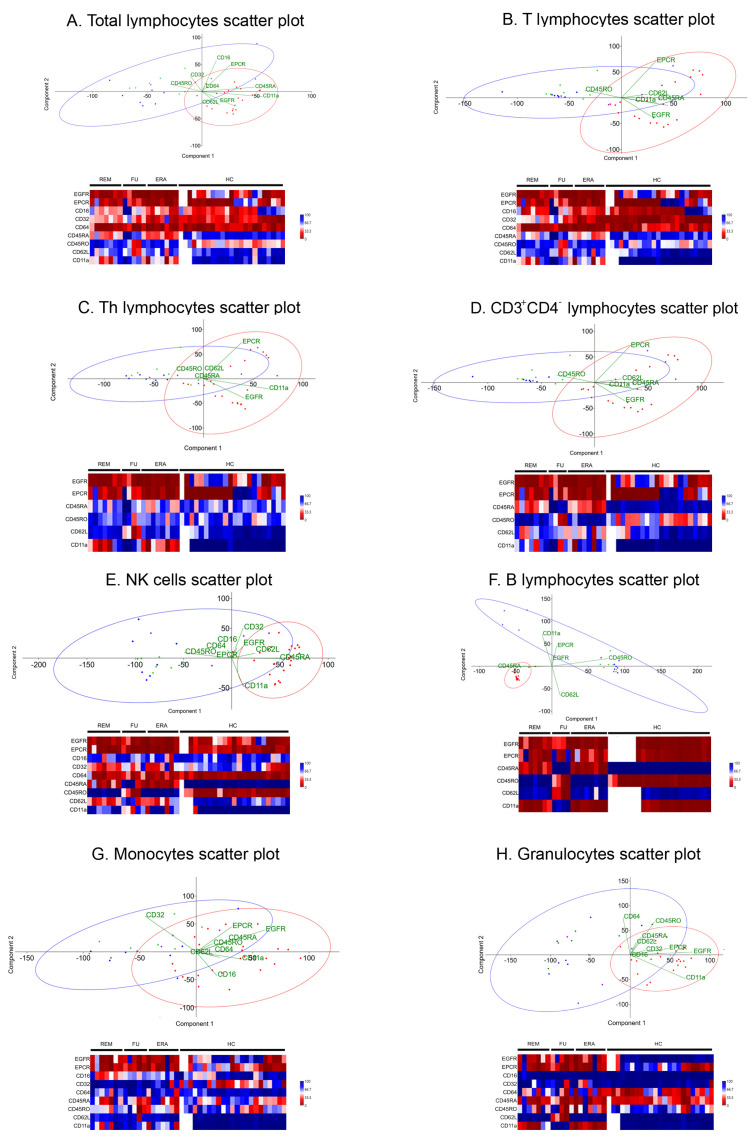
Analysis of leukocyte marker expression. Expression level of epidermal growth factor (EGFR), endothelial protein C receptor (EPCR), CD16, CD32, CD64, CD45RA, CD45RO, CD62L, and CD11a in total (**A**), T (**B**), and Th (**C**) lymphocytes, CD3^+^CD4^−^ lymphocytes (**D**), NK cells (**E**), B lymphocytes (**F**), monocytes (**G**), and granulocytes (**H**) in study subjects. XY scatter plot showing the principal component analysis (PCA), performed using 95% confidence ellipses for HC (in red) and RA patients (in blue). Matrix plot represents the correlation of the expression of the different leukocyte markers in HCs, ERA, flare-up (FU), and remission (REM) patients. Higher correlations in the matrix plot are shown in blue.

**Figure 4 ijms-24-12351-f004:**
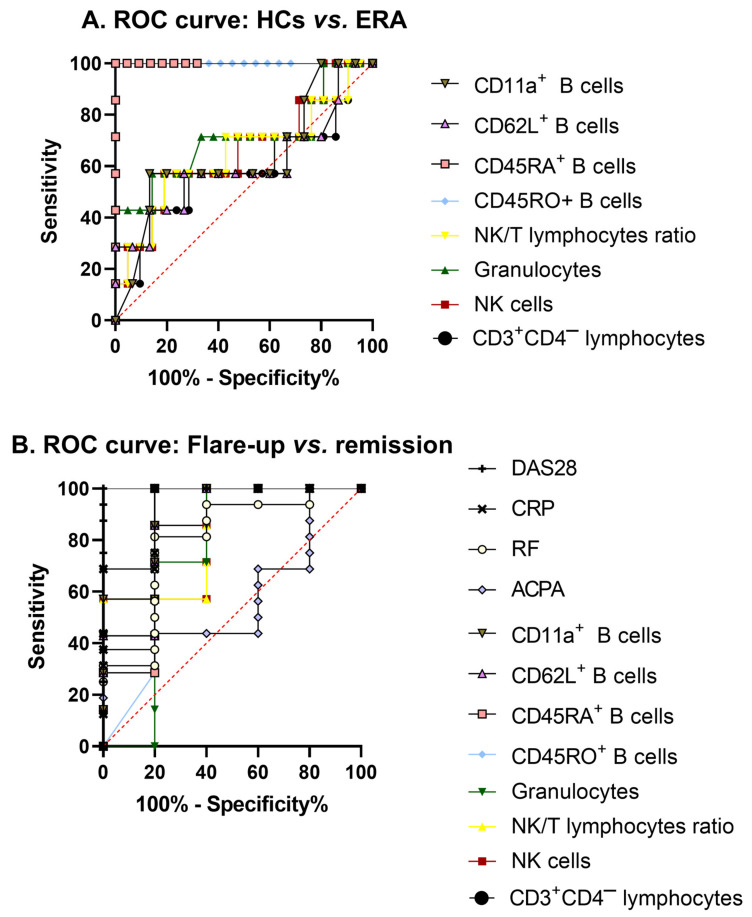
Receiver operating characteristic (ROC) curves of leukocyte antigens. Analysis of discrimination between HCs vs. ERA patients (**A**) and flare-up vs. remission patients (**B**) using CD3^+^CD4^−^ lymphocytes, NK cells, NK/T lymphocyte ratio, granulocytes, CD45RO^+^, CD45RA^+^, CD62L^+^, and CD11a^+^ B lymphocytes, ACPA, RF, CRP, and DAS28. *p*-values and the area under the curve (AUC) for each of the ROC curves were calculated using the Wilson/Brown method.

**Figure 5 ijms-24-12351-f005:**
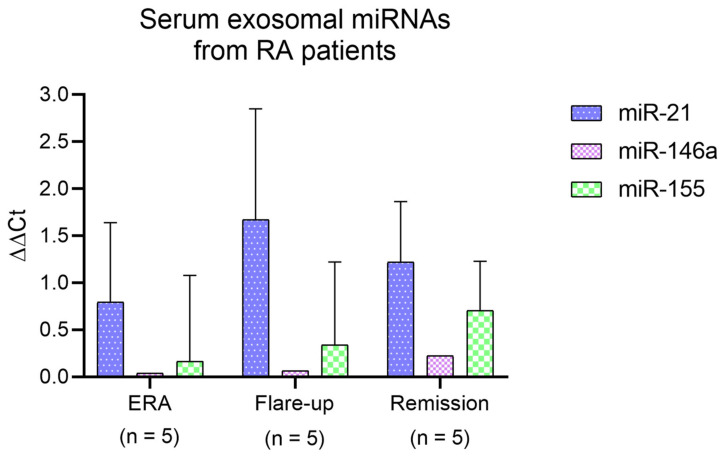
Analysis of exosomal miRNA (miR) levels. Serum exosomal miR-21, miR-146a, and miR-155 expression was measured (in ΔΔCt) in ERA, flare-up, and remission patients. Bars show mean. *p*-values were calculated using two-way ANOVA (Tukey’s test).

**Table 1 ijms-24-12351-t001:** Analysis of the treatments given to rheumatoid arthritis (RA) patients. Different synthetic, targeted synthetic, and biologic disease-modifying antirheumatic drugs (DMARDs) were given to early RA (ERA), flare-up, and remission patients. NSAIDs (non-steroidal anti-inflammatory drugs); JAK (Janus kinase); TNF (tumor necrosis factor); IL (interleukin).

Treatments for RA Patients			
	ERA	Flare-Up	Remission
Only methotrexate	7	-	-
NSAIDs or other drugs	3	-	-
JAK inhibitors	-	1	-
Methotrexate and anti-TNF-α antibodies	-	3	3
Methotrexate and anti-IL-6 receptor antibodies	-	-	3
Anti-TNF-α antibodies	-	1	2
Anti-IL-6 receptor antibodies	-	-	8

**Table 2 ijms-24-12351-t002:** Evaluation of clinical and analytical features of RA patients. Different clinical and analytical parameters were measured in healthy controls (HC), ERA, flare-up, and remission subjects. Results are expressed as mean ± standard error of the mean (SEM). *p*-values were calculated using an analysis of variance (ANOVA) (Fisher’s least significant difference (LSD) test) and Kruskal–Wallis test (uncorrected Dunn’s test). * Statistical differences. CRP (C reactive protein); RF (rheumatoid factor); ACPA (anti-citrullinated protein antibodies); DAS28 (disease activity score 28).

Clinical and Analytical Features of Study Subjects							
	Group				*p*-Values		
	HC	ERA	Flare-Up	Remission	ERA vs. Flare-Up	ERA vs. Remission	Flare-Up vs. Remission
n	39	10	5	16			
Ratio women/men	27:12	8:2	4:1	10:6			
Age (years)	53 ± 5.6	51 ± 4.3	51.6 ± 8.6	61.8 ± 3.7	0.58	0.07	0.41
Disease duration (months)	-	9.8 ± 6.3	87.4 ± 40.1	114 ± 39	0.03 *	0.0001 *	0.48
Painful joints	-	6.7 ± 1.3	6.6 ± 3.4	0.1 ± 0.1	0.89	<0.0001 *	0.0012 *
Swollen joints	-	3.2 ± 1.1	1.4 ± 0.2	0.1 ± 0.1	0.85	0.0006 *	0.0028 *
Level of discomfort (mm)	-	35.6 ± 6.3	53 ± 8.7	6.6 ± 1.5	0.39	0.0011 *	0.0003 *
Level of pain (mm)	-	36.4 ± 6.6	53.6 ± 9.5	6.7 ± 1.5	0.54	0.0007 *	0.0006 *
Evaluation of RA symptoms by the rheumatologist (mm)	-	37.4 ± 6	45.3 ± 9.5	3.8 ± 1	0.69	0.0002 *	0.0006 *
CRP (mg/L)	-	10.4 ± 3.4	7.9 ± 2.7	1.2 ± 0.3	0.77	0.0035 *	0.0069 *
RF (mg/L)	-	137.8 ± 67	173.8 ± 79.1	58.7 ± 19.6	0.28	0.38	0.053
ACPA (mg/L)	-	298.9 ± 148.7	791.8 ± 590.5	357.6 ± 123.4	0.28	0.59	0.46
DAS28	-	4.2 ± 0.4	3.8 ± 0.3	1.3 ± 0.1	0.31	<0.0001 *	<0.0001 *

**Table 3 ijms-24-12351-t003:** Cytokine profiles of serum samples. Serum cytokine concentrations are shown in pg/mL. Results expressed as mean ± SEM. Comparisons and *p*-values were performed/calculated for each cytokine, using ANOVA (Fisher’s LSD test) and the Kruskal–Wallis (uncorrected Dunn’s test) test. * Statistical differences. IFN (interferon).

Cytokines Measured in Sera Samples (pg/mL)										
	Group				*p*-Values					
	HC	ERA	FU	REM	HC vs. ERA	HC vs. FU	HC vs. REM	ERA vs. FU	ERA vs. REM	REM vs. FU
n	5	5	4	5						
IL-17A	33.3 ± 8.9	19.6 ± 12	20.4 ± 15.5	55.8 ± 7.2	0.3728	0.4284	0.1523	0.9590	0.0283 *	0.0410 *
IFN-γ	11.5 ± 2	11.5 ± 3.1	15 ± 1.5	13.9 ± 2.2	>0.9999	0.3260	0.4745	0.3260	0.4745	0.7506
TNF-α	11.2 ± 1.4	36.5 ± 8.1	24.4 ± 7.3	46.4 ± 18.5	0.0074 *	0.1197	0.0111 *	0.3317	0.8878	0.4022
IL-2	32.3 ± 3.1	51.3 ± 8.4	76.2 ± 22.7	55 ± 11	0.6640	0.0986	0.5333	0.5046	0.9960	0.6271
IL-6	12.3 ± 2.3	75.4 ± 36.5	31.4 ± 7.1	73.8 ± 26.1	0.0854	0.6351	0.0928	0.2824	0.9632	0.2996
IL-4	17 ± 3.2	28.9 ± 7.4	56.9 ± 24	36.9 ± 12.1	0.4459	0.0579	0.1665	0.2388	0.5345	0.5537
IL-10	4.6 ± 0.4	7.2 ± 2.9	11.5 ± 5.3	15 ± 6.5	0.3354	0.0963	0.0413 *	0.4504	0.2816	0.7945

**Table 4 ijms-24-12351-t004:** Cytokines released by peripheral blood mononuclear cells (PBMCs). Cytokines released by non-stimulated (**A**) and stimulated PBMCs (**B**). Results expressed as mean ± SEM. Comparisons and *p*-values were performed/calculated for each cytokine using ANOVA (Fisher’s LSD test) and the Kruskal–Wallis (uncorrected Dunn’s test) test. * Statistical differences. Ab (antibody).

A. Cytokines Measured in Non-Stimulated PBMC Supernatant (pg/mL)						
	Group			*p*-Values		
	HC	RA	RA + Ab	HC vs. RA	HC vs. RA + Ab	RA vs. RA + Ab
n	5	12	12			
IL-17A	3.7 ± 2.6	7.1 ± 1.9	4 ± 1.6	0.3582	0.9985	0.2299
IFN-γ	ND	ND	ND	0.1355	0.2690	0.5965
TNF-α	2.8 ± 0.6	4.2 ± 0.7	2.4 ± 0.5	0.1638	0.7260	0.0278 *
IL-2	0.4 ± 1.1	6.6 ± 3.1	1.2 ± 2	>0.9999	>0.9999	>0.9999
IL-6	1.6 ± 1.1	7.3 ± 2	8.7 ± 2.2	0.1171	0.0485	0.6003
IL-4	ND	1.5 ± 1.5	ND	0.0840	0.3493	0.3371
IL-10	2.4 ± 0.9	2.1 ± 0.4	1.9 ± 0.3	0.7339	0.9132	0.5701
**B. Cytokines Measured in Stimulated PBMC Supernatant (pg/mL)**						
	**Group**			***p*-Values**		
	**HC**	**RA**	**RA** **+ Ab**	**HC vs. RA**	**HC vs. RA + Ab**	**RA vs. RA + Ab**
n	5	12	12			
IL-17A	37.6 ± 12	58.3 ± 20.6	37.8 ± 10.5	0.9188	0.8285	0.6870
IFN-γ	1115 ± 595.1	276.9 ± 85.5	54.6 ± 16	0.0143 *	0.0031 *	0.3946
TNF-α	358.9 ± 181.2	109.9 ± 25.2	3.9 ± 1	0.0120 *	0.0006 *	0.1488
IL-2	1.3 ± 1.1	0.6 ± 0.3	1.9 ± 0.5	0.6623	0.4732	0.1561
IL-6	6839 ± 2265	15537± 5973	5337 ± 1543	0.8550	0.5829	0.6337
IL-4	ND	0.2 ± 0.9	0.3 ± 0.7	0.4299	0.1948	0.5084
IL-10	1494 ± 340.3	1021 ± 289.1	328.9 ± 77.5	0.2392	0.0064 *	0.0298

## Data Availability

The data presented in this study are available upon request from the corresponding author.

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
