# Peer review of "Analysis of Novel Immunological Biomarkers Related to Rheumatoid Arthritis Disease Severity"

_ijms, 2023, doi:10.3390/ijms241512351_

Round 1

Reviewer 1 Report

In the present study authors measured leukocyte surface antigens in patients with rheumatoid arthritis and healthy control subjects. The RA patients were divided into subgroups with early disease, patients during disease exacerbation and patients during remission.

 1)     The title does not correspond with the manuscript content. The patients were not followed-up in this study. They measured a lot of markers in three separate groups of RA patients and these markers may be considered as the markers of disease severity rather than progression.

2)     The information about treatment of patients included is missing. This information should be provided and the relationship between treatment and markers of interest should be analyzed.

3)     The number of RA patients included was very small, especially in specific subgroups. Was the required number of subjects calculated before the study?

4)     While describing the results, RA patients should first be compared to control subjects and then specific RA groups should be compared between them. This would be helpful in following the results.

5)     When signs of statistical significance are explained in table/figure legend, it should always be specified which group is used as the reference one.

6)     Lines 240-243: the discussion about effects of treatment is not supported by the data before this information is not presented in the manuscript.

7)     If the given marker is considered for clinical application, it would be reasonable to verify its diagnostic value vs. standard ones by methods such as ROC analysis or logistic regression analysis.

8)     Line 345: ethical aspects of cell culture studies are mentioned; what about agreement to perform the study in patients?

9)     Lines 407-408: details about the methods mentioned should be provided.

10) Section 4.7: more details about RT-qPCR should be provided including method used to measure RNA concentration and integrity, time and temperature of consecutive cycle phases and method of data analysis

11) Line 414: why miR-16 was used as the control?

12) Lines 421 and 427: why one-way and two-way ANOVA were used for different sets of samples?

Reviewer 2 Report

The Manuscript: „Analysis of leukocyte surface antigens as markers of rheumatoid arthritis disease progression’’ by Sandra Pascual-García and colleagues analyses and characterizes different leukocyte populations and their membrane expression of EGFR, EPCR, IgG Fc receptors, adhesion molecules and cell activation markers. The authors emphasize that the study of leukocyte surface markers could have potential as biomarkers of Rheumatoid Arthritis and their fluctuations could be related to the development of the characteristic pro-inflammatory environment. The manuscript is nicely written with adequate research of literature. After going through the manuscript, I have a couple of comments for the authors:

1.     Leukocyte surface antigens are known to be potential markers for RA disease progression, such as their accessibility through minimally invasive techniques and their ability to capture systemic immune changes. However, there are some challenges associated with these markers, including inter-individual variability, lack of standardized protocols, and potential confounding factors. Please discuss this point in the manuscript.

2.     I feel that the sample size of the study is small to generalize the statistical outcome. Was a sample size analysis performed?

3.     Discuss the potential clinical implications of using leukocyte surface antigens as markers for RA disease progression.

English is fine. Minor grammatical issues and some syntax errors need to  be fixed.

Reviewer 3 Report

- the last paragraph of the introduction should be rearranged in order to provide the general objective of the the study, instead of a summary of the methods.

- the first subsection of the methods should be about study design and population, with clear inclusion and exclusion criteria, which are not clear. The number of recruited patients then should be a result. 

- By the way, I recommend to place the methods after the introduction. 

- In the ethical statement, there is no mention about the informed consent about the participation in this study.

- Please, can you also clarify the study period.

- The definition of the healthy controls and their recruitment procedure is not clear at all. 

- In the results, more information about the specific therapy received by the study participants, perhaps by adding a new table, would be useful. 

- In the results and in figure 1, CTLs and Th cells should be clearly defined according to the immuno-phenotypical markers.

- Here, the terms of granulocytes is quite generic. Do you have information about the different types?

- Overall, the results are quite narrative. Some tables with numerical values and corresponding statistical significance would be useful; moreover, the main numerical findings could be inserted in the text as well.

- The colors use in table 2 is not completely clear, even if the authors mentioned that in the captions. I think that the authors may also rearrange this table in order to express the p-values. 

- Overall, the discussion sounds quite dispersive. I would recommend the authors to highlight at the beginning the main findings and, then, discuss them one by one. 

- I think the study limitations are more than one and recommend the authors to expand this part of the discussion.

- The conclusion is too long. I would suggest to list only the main take home messages clearly deriving from the present data. In the current form, it looks more a second abstract.

Minor-moderate editing.

Round 2

Reviewer 1 Report

The manuscript has been revised according to the reviewers' comments. I have no further concerns.

Reviewer 3 Report

Unfortunately, most of my comments were not appropriately addressed by the authors and some methodological aspects are definitely unclear. 

The general aim at the end of the introduction is not clearly stated. 

As said, the patients and methods section should be preferentially placed after the introduction, since it is essential to fully understand the results. 

The study design was not clearly and precisely stated, as previously recommended. 

The date of the IRB approval is not provided and the informed consent procedure is not fully described.

Despite the previous request, the authors has not clarified at all the recruitment of the controls, which are still clinically undefined. 

The immuno-phenotypic definition of lymphocyte subsets is questionable. For instance, the definition of CTLs as CD3+CD4- is not acceptable. The equipment does not justify this limitation, indeed FACS canto can easily perform a multi-parametric analysis. 

There are methodological concerns, which definitely impair the reliability of the results.

Round 3

Reviewer 3 Report

Unfortunately, I still have concerns. As regards, the IRB approval, it seems to be dated in 2019, but this study was prospective, even if cross-sectional. By the way, the study design and population is not clear yet, including the exact definition and recruitment of healthy controls, but not only. 

The authors state "Due to the flow cytometer used we could only measure 6 fluorescent parameters at a time. For thisreason, we decided to sacrifice 1 of the parameters (CD8), that in principle we could extrapolate through the analysis of CD3+CD4- cells". The "sacrifice" of CD8 is not acceptable and it is not true that the authors can "extrapolate" CD3+CD8+ cells as CD3+CD4- cells, since there are DNT and DPT cells which should be considered, even if there are a minor population. 

See comments above
